# Impact of Holder Materials on the Heating and Explosive Breakup of Two-Component Droplets

**Dmitry Antonov [1],\*, Jérôme Bellettre [2] , Dominique Tarlet [2], Patrizio Massoli [3] , Olga Vysokomornaya [1] and Maxim Piskunov [1]**

[1] National Research Tomsk Polytechnic University, Tomsk 634050, Russia; vysokomornaja@tpu.ru (O.V.); piskunovmv@tpu.ru (M.P.)

[2] Université de Nantes, Rue Christian Pauc, BP 50609, 44306 Nantes CEDEX 3, France; Jerome.bellettre@univ-nantes.fr (J.B.); dominique.tarlet@univ-nantes.fr (D.T.)

[3] Istituto Motori–Consiglio Nazionale delle Ricerche, Via Marconi 8, 80125 Napoli, Italy; p.massoli@im.cnr.it

\* Correspondence: antonovdv132@gmail.com; Tel.: +7-(913)-879-43-88

**Abstract:** The heating of two-component droplets and the following explosive breakup of those droplets have been extensively studied over the most recent years. These processes are of high interest, since they can significantly improve the performance of many technologies in fuel ignition, thermal and flame liquid treatment, heat carriers based on flue gases, vapors and water droplets, etc. Research throughout the world involves various schemes of droplet heating and supply (or, less frequently, injection) to heating chambers. The most popular scheme features the introduction of a two-component or multi-component droplet onto a holder into the heating chamber. In this research, we study how holder materials affect the conditions and integral characteristics of droplet heating and explosive breakup: heating time until boiling temperature; minimum temperature sufficient for droplet breakup; number and size of fragments in the resulting droplet aerosol, etc. Experiments involve droplets that are produced from flammable (oil) and non-flammable (water) components with significantly different thermophysical and optical properties, as well as boiling temperature and heat of vaporization. The most popular elements with the scientific community, such as ceramic, steel, aluminum, copper, and phosphorus rods, as well as a nichrome wire, serve as holders. We establish the roles of energy inflow from a holder to a droplet, and energy outflow in the opposite direction. We compare the holder results with a supporting thermocouple, recording the drop temperature under a heat transfer provided at 350°C. Finally, we forecast the conditions that are required for a significant improvement in the performance of thermal and flame water treatment through the explosive breakup of two-component droplets.

**Keywords:** two-component droplet; heating; evaporation; explosive breakup; disintegration; droplet holder material

---

## 1. Introduction

### 1.1. Motivation

To improve the thermal treatment of sewage and service water (in particular, in the form of an atomized flow) and to develop new, more effective technologies for it, we need to explore the physics of droplets of water solutions, slurries, and emulsions traveling through high-temperature gases. Their temperature exceeds 500 °C, and the most frequently used gases are hot air, fuel combustion products, and their mixtures. Unfortunately, there is still no theory of interconnected heat and mass transfer and phase transformations for such conditions. However, over the recent years, researchers have obtained

experimental results (e.g., [1–3]) that can become the premises for such a theory. No research findings on these processes have been published so far, because mathematical modeling becomes difficult for a large number of interfaces with highly nonlinear boundary conditions of rapid vaporization. Sazhin [4] outlined these difficulties in his review paper analyzing the reasons behind the slow development of models simulating the rapid heating and evaporation of droplets of fuels and emulsions based on them.

Experimental results [2] established that the leading droplets in a flow through hot gases significantly affect the heat exchange of the following droplets with the surrounding medium. Volkov et al. hypothesize [3] that, due to rapid vaporization, the first droplets considerably reduce the gas temperature in the front of all the following droplets. So, a thermal insulation of sorts is created for the following droplets in the form of a vapor curtain with a lower temperature, as compared to that of the gas medium in front of the first droplets. Until now, there have been no experimental or theoretical research findings on the thermal insulation of rapidly evaporating liquid droplets. It is important to obtain reliable experimental data and use them to develop adequate physical and mathematical models of heat and mass transfer. When analyzing the overview by Sazhin [4], we concluded that most likely, it is only possible to solve the formulated problem using optic techniques. Reliable information is necessary on temperature distributions in droplets of water and water-based solutions, slurries, and emulsions when rapidly heated. At the same time, we can infer from the findings by Sazhin [4] that the unsteady heating of a droplet has a significant impact on its lifetime. Under such conditions, the assumption of a constant temperature field of an evaporating and shrinking droplet cannot really be considered valid.

Snegirev [5] made attempts at analyzing the temperature gradient of an evaporating droplet to develop simplified mathematical models of phase transformations. He formulated dimensionless criteria to estimate the temperature gradient within a droplet, and its impact on liquid evaporation rate. However, no experimental data to support the reliability of such estimates have been published so far. The task also becomes more complex, because the research needs to be done at relatively high temperatures of the gas medium (over 500 °C). Vysokomornaya et al. [6] show that traditional evaporation models also known as kinetic and diffusion models based on the assumed dominating process [7–9] provide a good agreement between the theoretical research and experimental data only at moderate gas medium temperatures (under 500 °C).

The evaporation of liquids also remains understudied because its intensity depends on the surface temperature of the phase transition and the concentration of liquid vapors in the small-size area next to the interface region. Diffusion and heat transfer in this area are the main drivers of evaporation. Experimental data on the main characteristics of heat and mass transfer near the surface of evaporating droplets are not yet published.

In the considered research area, an unsolved problem is the need to provide the controlled conditions for the crushing droplets due to overheating and micro-explosions. The use of the controlled effects of explosive breakup will solve a number of problems in the areas of unmixed and mixed fuels: combustion stabilization throughout the combustion chamber, reducing heating and ignition costs, increasing calorific value, reducing anthropogenic emissions, improving rheological properties, etc. [10]. These impact on the research in this area.

Explosive breakup of water emulsion and slurry droplets in a high-temperature gas environment was studied experimentally [11–13]. For the explosive disintegration of heterogeneous droplets to happen, the temperature at the interface must reach that of water boiling. A non-contact method, planar laser-induced fluorescence (PLIF), made it possible to establish that the temperature near this interface reached 100–120 °C before disintegration. The authors determined the threshold temperatures at the onset of this effect for a group of solid and liquid organic additives (slurry and emulsion components). As a result of the explosive fragmentation of multi-component droplets, the evaporation surface area increases up to 15 times. It is important to expand the experimental database with the evaporation characteristics of typical sewage and service water compositions to improve their treatment.

The most popular approach to the experimental research into the breakup of boiling liquid, solution, emulsion, and slurry droplets is placing them on a holder into a heated gas flow (e.g., [11–16]). Some setups do not include different holders [14–16] or use substrates [17]. Each of these recording schemes has its own strengths and weaknesses [10]. In terms of the costs and difficulty of the experiment, as well as the reliability of the recording procedure, a holder seems to be the most rational option. However, the choice of the holder material for the fragmentation of boiling droplets of liquids, solutions, emulsions, and slurries is yet to be studied. It is of interest to study how this factor affects the heating and disintegration of typical two-component droplets using a large group of popular materials. A solution to this problem is of principal importance for the development of high-potential gas-vapor-droplet technologies considered in [18–27]. Vershinina et al. [10] established the impact of the holder material on the ignition of fuel slurry droplets. They show that there are two temperature ranges. Above 600 °C, the impact of the holder material is negligible, while below 600 °C, the properties of the said material have a significant influence on heat transfer. It is important to make such estimates for a group of promising two-component droplets.

Another approach [28,29] consists of suspending the emulsion drop onto a thermocouple junction, enabling to measure its temperature during its evaporation under heating. Its results are compared to the present study, since the experimental conditions are similar concerning both emulsion properties and heat source temperature.

The purpose of this work is to study experimentally how the holder material affects the heating, evaporation, and explosive breakup of two-component droplets.

### 1.2. Review of Time Ranges of Droplet Breakup through Microexplosion

This subsection presents a review of the time ranges of droplet breakup through microexplosion. We considered the experimental results of microexplosion times published in studies [30–33]. Table 1 contains the main suitable data from these papers.

**Table 1.** Review of time ranges of droplet microexplosion established in the experiments accomplished by using different experimental techniques.

| Article | Components | Material of Holder | Range of Two-Component Droplet Breakup Times | Experimental Setup |
|---|---|---|---|---|
| [30] | Water + n-dodecane Water + n-tetradecane Droplet size $V_d$ = 5–15 μm | **Quartz fiber $D$ = 0.25 mm** | On the holder (0.22–0.85 s) During fall (0.25–0.95 s) | A droplet is placed on the holder inside the combustion chamber with a temperature of 30 °C. After that, the droplet ignites by an electrically heated wire. The temperature of the droplet is measured by the Pt–PtRh thermocouple. A video camera records the microexplosion process. The fall process lies in the simultaneous motion of the chamber and the droplet during ~ 1 s. |
| [31] | Pure bio-oils $D_0$ = 1.12 mm Pure bio-oils $D_0$ = 1.08 mm | **A droplet is fixed on a thermocouple junction (K-type)** | $t$~**7s** ($T_a$ = 300 °C) $t$~**4s** ($T_a$ = 500 °C) | A droplet is fixed on a thermocouple. By using a linear module, it is introduced into the space between two plates heated by electricity. |
| [32] | Ethanol + Jet A-1; $D_0$ = 2 mcl | **Quartz holder $D$ = 0.2 mm** | (1.5–2.3 s) | By using a dispenser, a droplet is placed on a holder. The droplet ignites by using a nichrome wire. The process under study is recorded by a high-speed video camera. |
| [33] | Heptane C7H16 + Hexadecane C16H34 | **Without holder** | (170–205 ms) | A device is applied to collide two droplets of the required size, and to form a two-component droplet. The droplet moves through the combustion chamber heated up to 1050 °C. High-speed video recording allows the determination of droplet lifetimes and their breakup times. In addition, as a comparison, the experiments are performed with the preliminary formed two-component droplets. |
| [28,29] | Sunflower oil, distilled water, non-ionic surfactant SPAN 83 | **K-type thermocouple (Nickel–Chromium, Nickel–Alumel)** | (0.9–1.3 s) | A bare K-type thermocouple (wire diameter 76.2 μm) is heated from below by the means of a highly resistive coil with its asymptotic temperature of 350 °C. The emulsion drop is maintained on the thermocouple junction by interfacial tension. The thermocouple signal is acquired by an oscilloscope, and the shadowgraph frames are visualized using a high speed camera (10,000 fps). |

An analysis of data presented in Table 1 enables the conclusion that when using the different holders examined in the study, the explosive breakup times of the droplets are the upper estimates of

the actual values in the practical applications. Therefore, the values presented in the research can be used to predict the maximum possible times of heating until droplet breakup.

In addition, the challenging task is to determine the times of the heating until breakup, the complete volatilization of impurities, and the ignition of different heterogeneous droplets at free fall, i.e., without the holder [33]. Nowadays, such experiments are labor-intensive and expensive. Thus, the various holders, including those used in the study, will be employed for a rather long time. In such a situation, a utilization of the research results can help to predict differences between droplet-heating characteristics for a large group of experimental studies performed or planned to be carried out by using various holders.

## 2. Experimental Setup and Procedure

### 2.1. Components of Two-Component Droplets and their Production Procedure

The experimental research featured two components: water (with a specialized dye–fluorophore Rhodamine B) and transformer oil. The main properties of components are presented in Table 2. The component concentrations were varied over a wide range as per recommendations. The fusion of these liquids resulted in a two-component droplet. The Rhodamine B dye was used to control water temperature in a two-component droplet, similar to the methods used in [16,17]. It was important to provide the same conditions as those used in experiments [16,17] in order to extrapolate the experimental results to various schemes of energy supply to a two-component droplet. Unsteady and inhomogeneous temperature fields of droplets obtained experimentally were in good agreement with the results from [16,17]. Therefore, further analysis will focus on the impact of holder materials on heating.

**Table 2.** Main properties of the liquids under study.

| Component | Thermal Physical Properties | Kinematic Viscosity, $m^2$/s | Surface Tension, N/m | Boiling Temperature, °C | Heat of Vaporization, MJ/kg |
|---|---|---|---|---|---|
| **Transformer Oil** | $\rho = 877$ kg/$m^3$, $\lambda = 0.12$ W/(m·°C), $C = 1670$ J/(kg·°C), $a = 8 \cdot 10^{-8}$ $m^2$/s | $22 \cdot 10^{-6}$ $m^2$/s at 20 °C, $0.295 \cdot 10^{-6}$ $m^2$/s at 100 °C | $26.15 \cdot 10^{-3}$ | 320 | 0.209 |
| **Water** | $\rho = 1000$ kg/$m^3$, $\lambda = 0.6$ W/(m·°C), $C = 4200$ J/(kg·°C), $a = 14 \cdot 10^{-8}$ $m^2$/s | $1.006 \cdot 10^{-6}$ $m^2$/s at 20 °C, $2.56 \cdot 10^{-6}$ $m^2$/s at 100 °C | $72.86 \cdot 10^{-3}$ | 100 | 2.258 |
| **Sunflower Oil** | $\rho = 865$ kg/$m^3$, $\lambda = 0.165$ W/(m·°C), $C = 2500$ J/(kg·°C) | $6.03 \cdot 10^{-5}$ $m^2$/s at 25 °C | $33.7 \cdot 10^{-3}$ | 225 | 0.21 |

### 2.2. Holder Materials

Copper, aluminum, ceramics, steel, nichrome, and phosphorus were the main materials used to produce the holders for two-component droplets under study, since these materials have a wide range of values of thermal and physical characteristics (Table 3).

**Table 3.** Thermal and physical characteristics of holder materials (average values for the temperature range of 200–500 °C in line with the experiments).

| Material | $\lambda$, W/(m·°C) | $C$, J/(kg·°C) | $\rho$, kg/$m^3$ | $a \cdot 10^6$, $m^2$/s |
|---|---|---|---|---|
| **Copper** | 376.86 | 416.12 | 8770.31 | 103.4 |
| **Aluminum** | 229.56 | 1044.76 | 2642.526 | 83.62 |
| **Ceramic** | 1.4 | 770 | 2355 | 0.772 |
| **Steel** | 42.8 | 561.8 | 7723 | 9.912 |
| **Nichrome** | 22.5 | 460 | 8660 | 5.648 |
| **Phosphorus** | 0.236 | 23.82 | 1820 | 5.444 |

Figure 1 shows the images of the holders used in the experiments: 1—ceramic; 2—steel tube; 3—aluminum; 4—copper; 5—nichrome; 6—phosphorus; 7—steel. When using each of the holders,

we measured its contact area with the droplet. The largest holder/droplet contact area was found in the experiments with ceramic and aluminum rods, and the smallest one, with a nichrome wire. The contact surface area of the droplet and holder surface ($S_h$) mainly depended on the droplet radius ($R_d$) and the holder size ($d_h$), considering that the radius of an evaporating droplet decreases nonlinearly. The contact area was calculated using the formula from Figure 1.

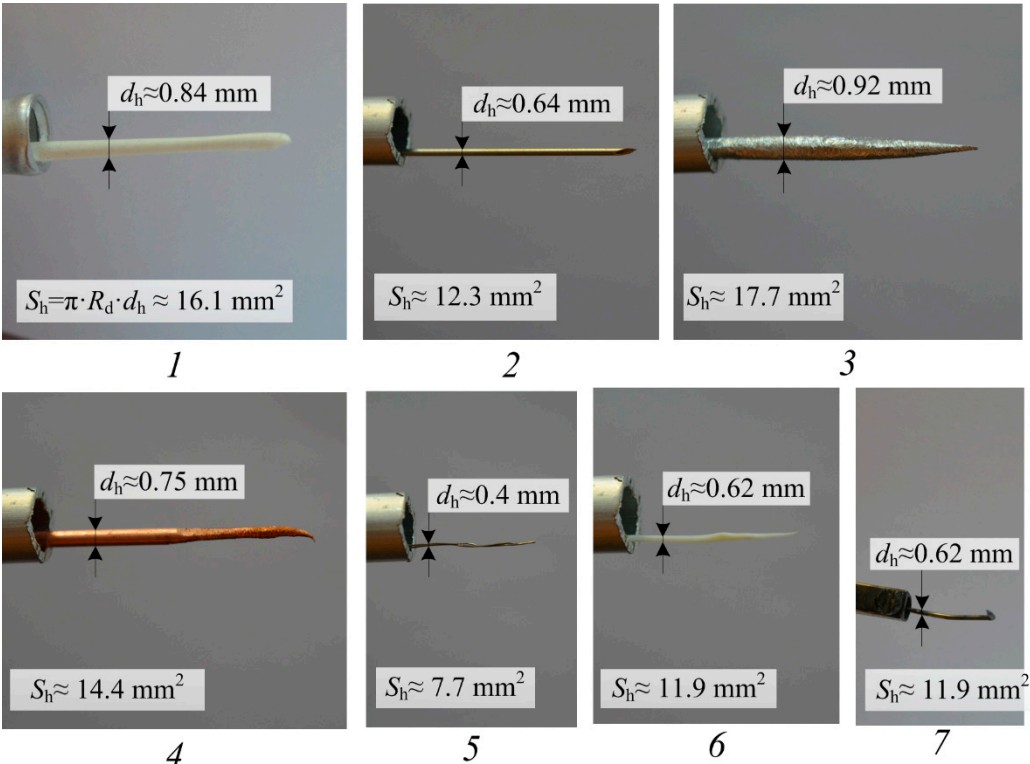

**Figure 1.** Appearance of holders used in the experiments (their size and contact area with a droplet are specified) 1—ceramic; 2—steel tube; 3—aluminum; 4—copper; 5—nichrome; 6—phosphorus; 7—steel.

The holders were chosen to provide similar schemes of droplet fixation and contact surface. This made it possible to record in the experiments quite a similar geometry for the contact line (interface) between the liquid component and the holder. Under such conditions, the heating or cooling rates of a droplet mostly depended on the thermal and physical properties of the holder material (Table 3). The results of this analysis are presented further.

### 2.3. Methods for Studying the Disintegration of Boiling Droplets

Figure 2a shows the two-component droplet heating scheme at convective heating, as well as Figure 2b illustrates the actual photo of the two-component droplet during experiments. The Leister CH 6060 hot air blower (air velocity 0.5–5 m/s) (LEISTER Technologies AG, Switzerland) and a Leister LE 5000 HT air heater (temperature range 20–1000 °C) were used as a heating system, generating the necessary parameters of the flow of high-temperature gases (flow rate $U_a$ and temperature $T_a$). The flow of high-temperature gases was formed in a hollow transparent cylindrical channel (internal diameter 0.1 m, wall thickness 2 mm). A two-component droplet was placed on the holders under study (Figure 1), which were introduced into the flow of high-temperature gases using a motorized coordinate device (motorized manipulator).

We recorded the heating, boiling, and disintegration of two-component droplets by a high-speed video camera. The recordings were processed using the Tema Automotive and ActualFlow software packages for the continuous tracking of moving objects. In the course of processing, we determined the initial droplet radius $R_d$ and the total liquid evaporation surface area, $S$. The video recordings

were processed in two stages. At first, we tracked how the frontal cross-sectional area $S_m$ of an evaporating and deforming droplet changed until it finally broke up (Figure 2c). Using the Airbag and Advanced Airbag tracking algorithms, we observed the changes in the shape of an evaporating droplet. After that, the frontal cross-sectional area of a droplet was calculated, and the curves $S_m(t)$ were plotted. The droplet was assumed to be spherical and its frontal cross-sectional area to be a circle. Using the formula $R_d = (S_m/\pi)^{0.5}$, we calculated the average droplet radius $R_d$. The errors of the $R_d$ calculation did not exceed 2.5%. After that, the total area of the droplet evaporation surface was calculated using the formula $S = 4\pi R_d^2$.

Video recordings following the explosive breakup of the heterogeneous droplet into separate smaller fragments were analyzed at the second stage. A polydispersed aerosol was usually formed. The shadow image was analyzed using the Actual Flow software to determine the location, boundaries, and dimensions of separate droplets. Median, Low Pass, and Average software filters were used to screen off the noises, and Laplace Edge Detection was used to determine the boundaries of droplet surface. When determining the droplet dimensions, we applied the Bubble Identification algorithm. The error of the $R_d$ calculation using this approach was under 3%.

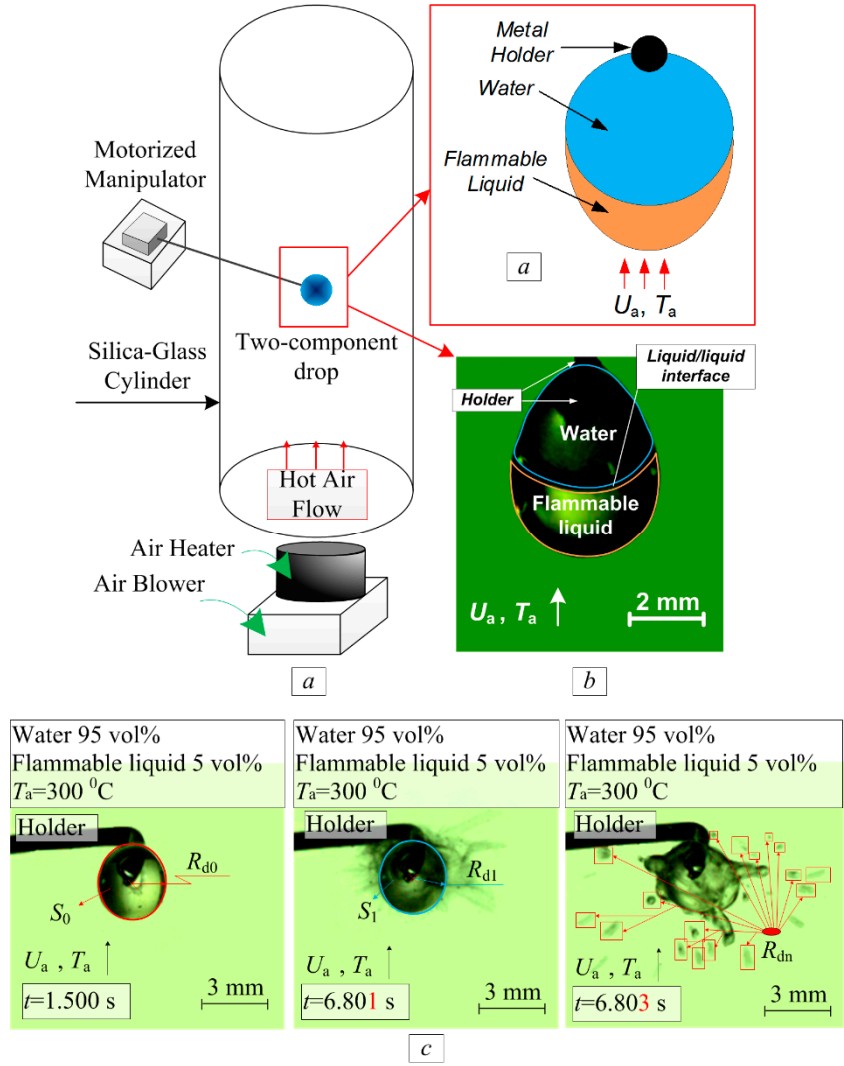

**Figure 2.** Methods for studying the disintegration of boiling droplets: (**a**): The scheme of registration of the heating process of two-component droplet; (**b**): Photo of the formed two-component droplet; (**c**): Scheme of recording heated droplet breakup and aerosol generation.

To obtain the droplet size distributions, all of the droplets were classified into *m* groups. We then determined the number of droplets *n* and the average droplet size $R_{dn}$ in each group. The liquid evaporation surface for the droplets in each group was calculated using the formula $S_n = n \cdot 4\pi R_{dn}^2$. As the final step, we calculated the overall evaporation surface area: $S = S_{n(1)} + S_{n(2)} + \ldots + S_{n(m)}$.

### 2.4. Main Registered Parameters and Tolerances

Table 4 presents the parameters recorded in the experiments, and the systematic errors of the measurement tools. The next section outlines the random errors calculated as part of statistical analysis of the results in a series of experiments.

**Table 4.** Main registered parameters and tolerances.

| Physical Magnitude | Droplet Volume ($V_d$) | Droplet Radius ($R_d$) | Temperature Inside the Droplet ($T_d$) | Two-Component Droplet Breakup Times ($\tau$) and Lifetimes ($\tau_h$) | Air Temperature ($T_a$) | Air flow Velocity ($U_a$) |
|---|---|---|---|---|---|---|
| **Measurement Tool/Technique** | Finnpipette Novus dispensers | High-speed cameras Phantom Miro M310 and Photron Fastcam SA1, Tema Automotive software | Planar Laser Induced Fluorescence (PLIF) | High-speed cameras Phantom Miro M310, Photron Fastcam SA1, and Phantom V 411, Tema Automotive software | Temperature meter (IT-8) | Particle Image Velocimetry (PIV) |
| **Systematic Errors** | $\pm 0.05\ \mu L$ | $\leq 4\%$ | $\pm 1.5$–$2\ °C$ | $\leq 4\%$ | $\pm(0.2 + 0.001T)\ °C$ | $\pm 2\%$ |

## 3. Results and Discussion

### 3.1. Droplet Disintegration Regimes

In this section, we present typical video frames showing the heating and disintegration of two-component droplets on holders made of different materials (Figure 3). The smallest droplets were formed in the cases when holders were made of materials with low thermal diffusivity (ceramics and phosphorus).

On a copper holder, the droplets did not reach the temperature sufficient for an explosive breakup, but gradually evaporated in a wide range of the main parameters: air temperature and component concentration. Explosive breakup was only observed at 450 °C, and with a 50/50 concentration of the flammable and non-flammable components. The heating time before disintegration was 25–30 s, most likely because copper is good at removing heat from the droplet. Copper holders have the highest thermal diffusivity (around $117 \cdot 10^{-6}\ m^2/s$). Droplets do not reach the conditions of micro-explosion (even boiling is not observed). The video frames of the experiments only showed monotonous evaporation.

Aluminum, however, with its thermal diffusivity of $90 \cdot 10^{-6}\ m^2/s$ (close to that of copper), provided quite a stable explosive breakup. The heating time until explosive breakup was longer than with the other holders, except for copper. The experiments established two factors prolonging the time of two-component droplet disintegration: heat removal from a droplet due to relatively high thermal diffusivity, formation of thermal stresses, nucleation, and growth of bubbles with high pressure in a droplet, which disintegrates to form smog or mist.

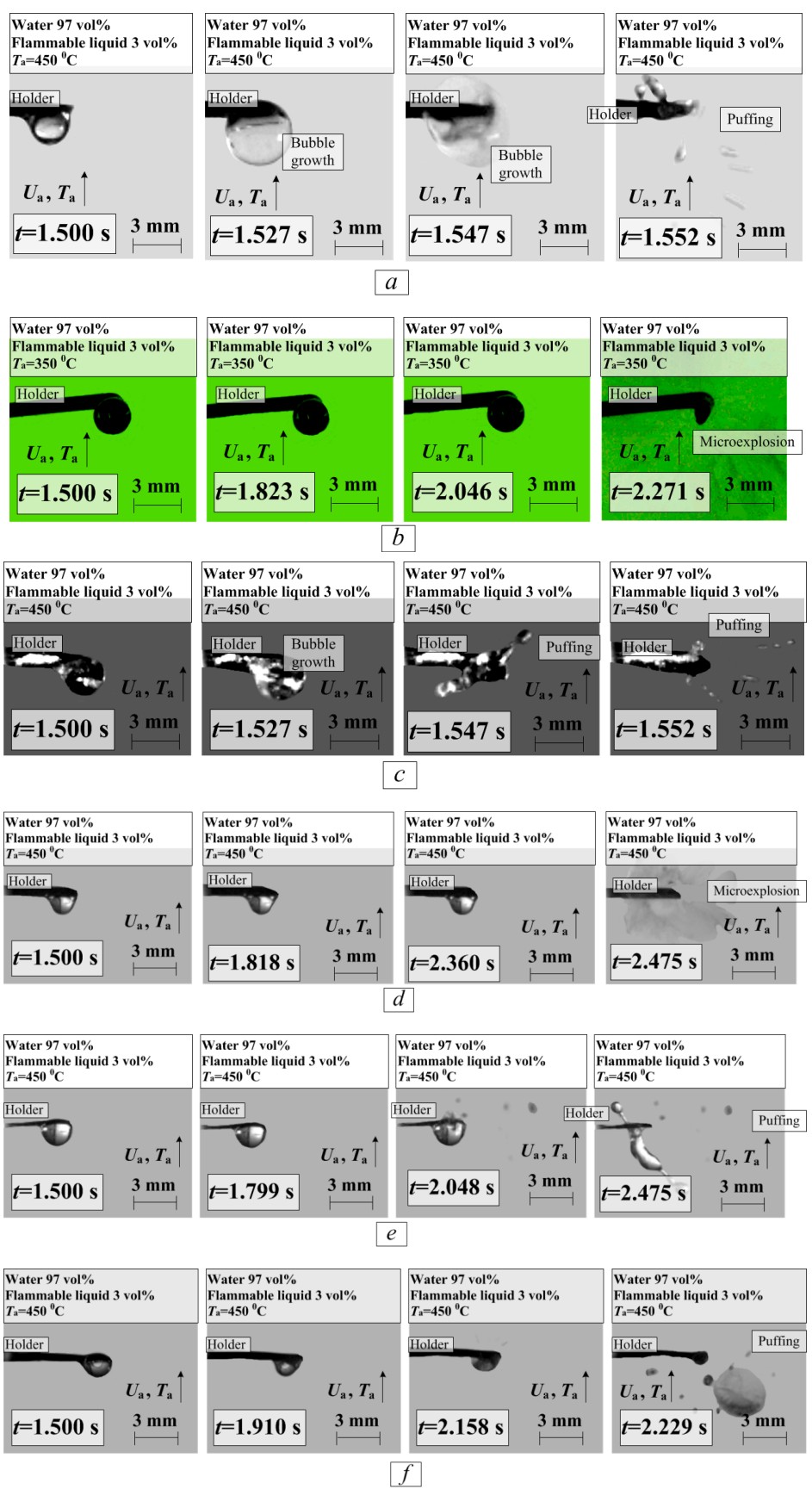

**Figure 3.** Snapshots of breakup or fragmentation on various holders: (**a**): ceramics; (**b**): steel; (**c**): aluminum; (**d**): steel tube; (**e**): nichrome; (**f**): phosphorus.

### 3.2. Impact of Key Factors

Figure 4 shows the times of two-component droplet disintegration vs gas medium temperature. The longest disintegration times were observed when using an aluminum rod as a holder for a two-component droplet. This results from the above-described possible rapid heat removal from a droplet due to high thermal diffusivity of aluminum. This plot also shows that the times of the two-component droplet disintegration decreased rapidly with an increase in the temperature. Such a pattern is typical of each holder material under study.

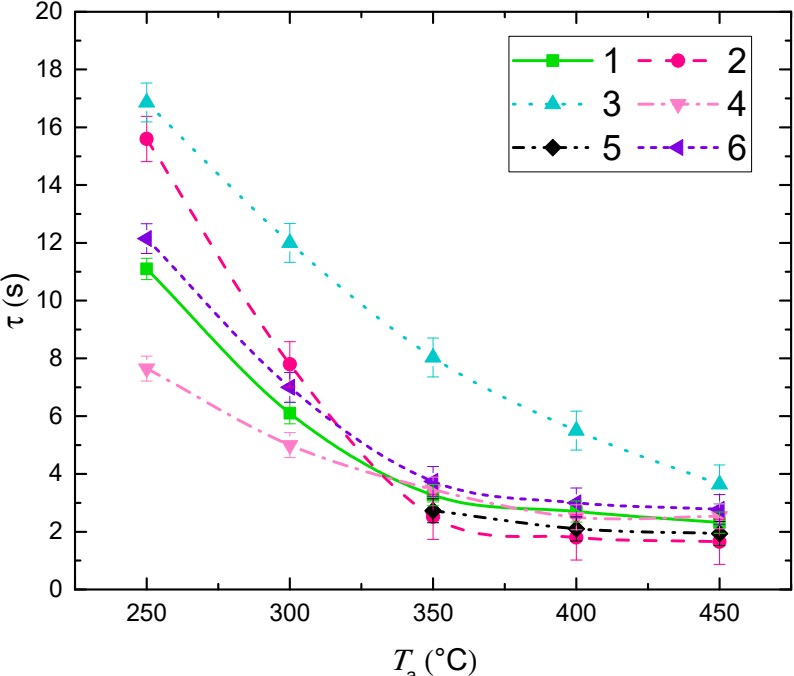

**Figure 4.** Two-component droplet heating times until explosive breakup vs gas medium temperature with various holders: 1—ceramic; 2—steel; 3—aluminum; 4—steel tube; 5—nichrome; 6—phosphorus. The value $T_a \approx 250\ ^\circ$C is the threshold for explosive breakup.

The temperatures at which the main experimental studies were performed to observe explosive breakup, ranged from 250 °C to 450 °C. The optimal temperature that steadily provided an explosive droplet breakup within a short time was 350 °C. Below that, the disintegration times increased non-linearly, and above that, they remained practically the same. Further increase in the $T_a$ is redundant and unpractical for water treatment and other energy-consuming applications.

Apart from temperature functions, the plots of two-component droplet disintegration times vs flammable component concentration were among the key ones. The resulting functions are highly non-linear, which suggests a significant impact of the flammable liquid concentration on a group of interconnected processes promoting the breakup of the initial two-component droplets (Figure 5).

These results can be compared to the recorded lifetime of the water-sunflower oil emulsion drop, supported by a K-type wire thermocouple [28,29] under the asymptotic temperature of 350 °C. It ranges from 0.9 to 1.3 s, close to the same order of magnitude. This result confirms that reproducibility of lifetimes can be obtained under the same conditions of heat transfer that are mainly determined by the temperature of the heat source. The emulsion temperature acquired by the thermocouple steadily increased until the boiling point of oil, which is less than 250°C (see Table 1).

Figure 5 shows the maximum droplet heating times until breakup with equal relative fractions of the flammable and non-flammable components. This stems from a set of factors and processes that are opposite in terms of their impact. Due to the higher heat capacity and the vaporization heat, water heats up rather slowly as compared to oil, but the thermal conductivity and thermal diffusivity of

the latter are several times lower than those of water. Therefore, under identical heating and equal component concentrations, these factors counterbalance each other. As a result, a two-component droplet is heated more slowly until it reaches the conditions of explosive breakup. Moreover, with a low proportion of water in a droplet, the film of the flammable component is thick, and it is heated faster than water. Thus, the heating times of the initial (parent) droplet until breakup are minimum. With the highest possible fraction of water and lowest fraction of the flammable component, the trend changes. A thin film of the flammable component is heated fast and locally overheats the near-surface water layer. This is enough for bubble nucleation at the interface and the explosive breakup of the initial droplet.

Moreover, the results shown Figure 5 can be compared to the emulsion drop lifetime of 0.9 to 1.3 s obtained using 70 vol % sunflower oil. The lifetime is close to the same order of magnitude. It confirms that reproducibility is not only a function of the heat source temperature, but also that of the emulsion properties.

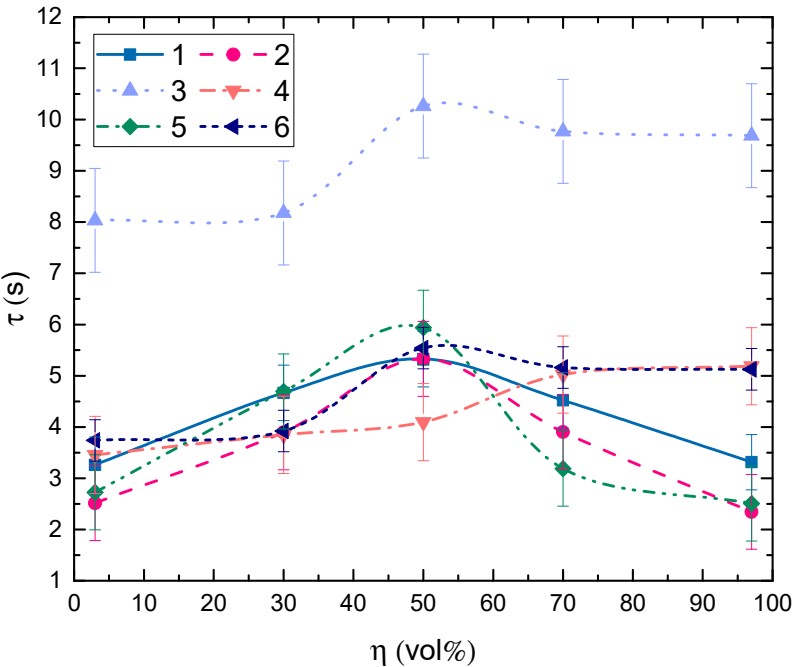

**Figure 5.** Droplet breakup times vs. relative mass fraction of the flammable component with various holders: 1—ceramic; 2—steel; 3—aluminum; 4—steel tube; 5—nichrome; 6—phosphorus.

### 3.3. Droplet Disintegration Outcomes

By analyzing the outcomes of the explosive breakup (Figure 6) of a two-component droplet, we have established that hollow steel and phosphorus tubes as well as a nichrome wire used as a holder yield droplet aerosols with a maximum quantity of small fragments. The liquid evaporation surface area increased more than 40 times under such conditions.

In the experiments with an aluminum holder, the evaporation surface area increased massively with the growing concentration of the flammable component. This results from the longer droplet heating time (Figure 5). The longer the period of droplet heating until breakup, the greater the volume of the two liquids that is heated to high temperatures. The droplet broke up into a greater number of fragments, which boiled and disintegrated in the process, into even smaller droplets. Presumably, the chain-like breakup of droplet aerosols may potentially intensify.

For a phosphorus holder, on the contrary, a low concentration of flammable liquid provides the largest evaporation surface area of droplets, most likely due to water boiling that is in contact with the holder surface. Since a phosphorus rod removes very little energy from the droplet, almost all the energy that is supplied is spent on heating the liquid components. The explosive breakup

occurred when the liquid–liquid interface was heated to water boiling temperature. A thin flammable component film quickly reached high temperatures, and so did a thin water layer at the interface, which was recorded in the experiments.

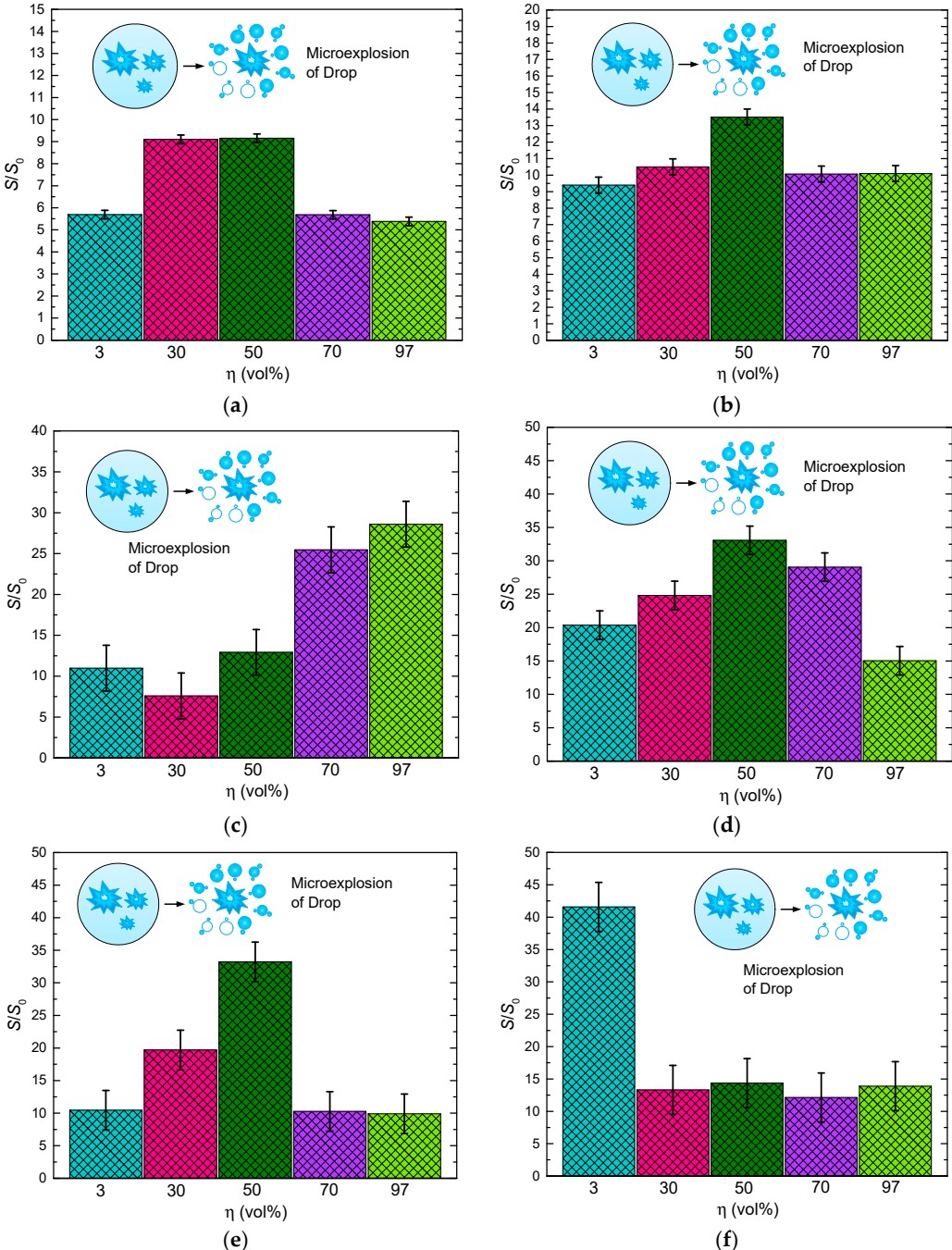

**Figure 6.** Ratios of surface areas of small droplets formed after the breakup of two-component droplets to their initial areas depending on the concentration of flammable liquid (oil) on various holders: (**a**): ceramics; (**b**):steel; (**c**): aluminum; (**d**): steel tube; (**e**): nichrome; (**f**): phosphorus.

For the other holder materials, the emerging droplets had the largest evaporation surface areas with 50/50 component concentrations. With this concentration of the flammable liquid, the breakup times were the longest. Therefore, a droplet has more time to form the temperature stresses and nucleation sites of vapor bubbles, i.e., to reach the temperatures sufficient for rapid vaporization near the inner water–flammable liquid interface.

A literature analysis shows that it is possible to significantly reduce the size of the droplets of various liquids (respectively, to increase $S/S_0$), due to several mechanisms. The most common are the following: the impact of the droplets between themselves; the interaction of droplets with an obstacle; the acceleration of droplets to the conditions under which they lose their stability and are significantly transformed; micro-explosive crushing due to overheating.

In Figure 7, we added the results of additional test experiments (carried out in accordance with the methods [34,35]) with droplets of oil–water emulsions (50% transformer oil, 50% water; 50% castor oil, 50% water). The choice of oils is due to the fact that we worked with transformers when studying micro-explosive effects (Figure 6), and the viscosity, density, and surface tension of castor oil are significantly different from transformer oil. Figure 7 shows that with an increase of the speed and size of the colliding drops, it is possible to ensure a multiple increase of the ratio $S/S_0$. The values of this parameter grow especially on a large-scale with temperature increasing, since the surface tension and viscosity of liquids decrease. At the same time, the scale of growth of this ratio correlates well with Figure 6 at micro-explosive decay. These results are the basis for the formulation of the hypothesis that the combination of the effects of droplet collisions and their overheating will increase the $S/S_0$ ratio by a 100 times or even more. Thus, it is possible to provide a significant increase in the efficiency of modern technologies of secondary grinding of droplets.

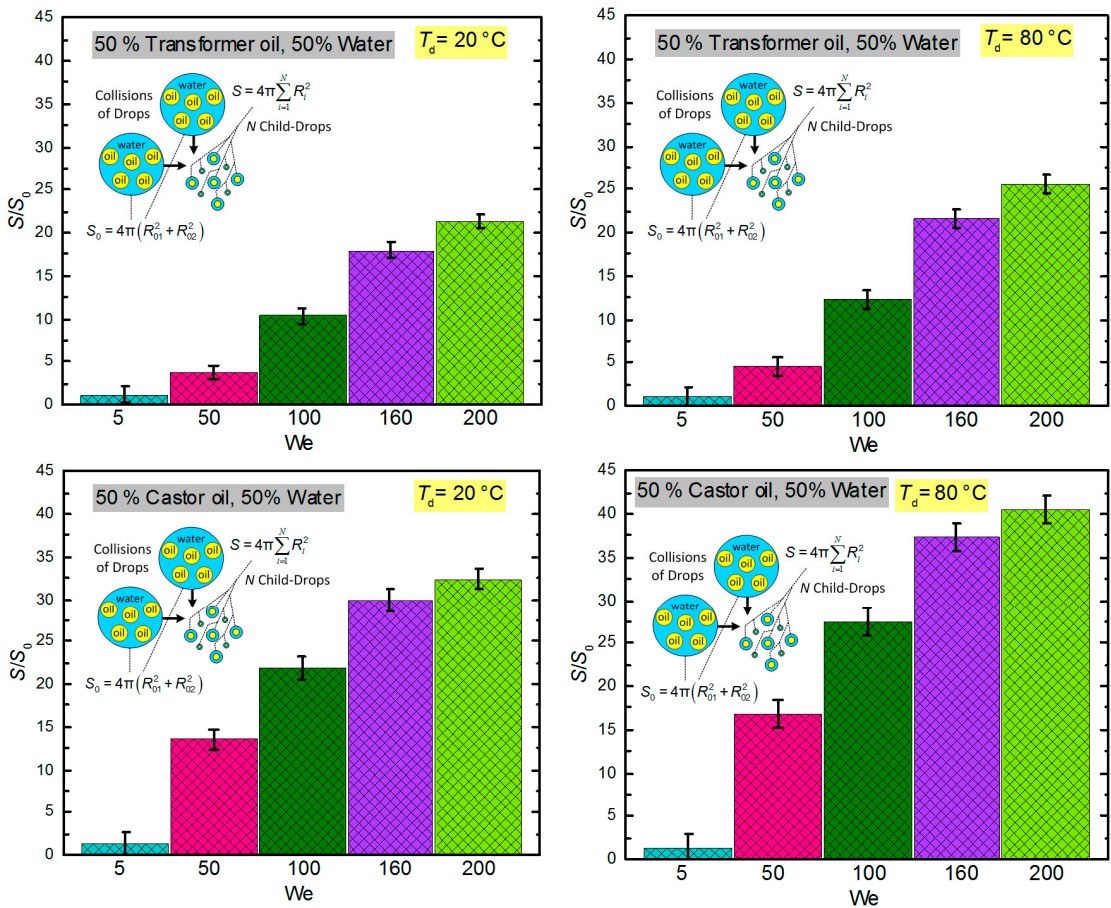

**Figure 7.** The results of additional test experiments (carried out in accordance with the methods [34,35]) with droplets of oil–water emulsions.

### 3.4. Generalization of Research Findings

The experimental results made it possible to determine how the holder material affects the heating of multi-component droplets, and to discover that in some cases, additional heating of such droplets due to their contact with the holder intensifies their explosive breakup. A major role belongs to the

direct contact of the non-flammable component (water) with the heated holder surface. Moreover, the experiments have established that the heating of the two-component droplets is more rapid if the core of the droplet is made of water, and the envelope, of oil. This happens because oil has a high absorption ability, and less energy is spent on its evaporation. Therefore, oil is heated faster than water, although water has a higher thermal conductivity and diffusivity than oil.

To demonstrate the heating conditions of multi-component droplets on a holder, we have developed a simplified one-dimensional mathematical model of heat transfer in the holder–two-component droplet system (Figure 8) similar to the one in [36]. This model determined the temperature variation trends for the holder, core and envelope of a droplet (Figure 9). Signature domains are divided by different colors in Figure 8. Similarly to the model in [36], we took into account the droplet heating through both thermal conductivity and radiation absorption, according to the Beer–Lambert–Bouguer law. At a first approximation, we used a one-dimensional statement to evaluate the variation of the vertical temperature profile, as shown in Figure 8. From the analysis of the temperature fields established in experiments [16,17], we can conclude that highly unsteady and inhomogeneous temperature profiles, which further determine the intensity of droplet breakup, are formed in such sections.

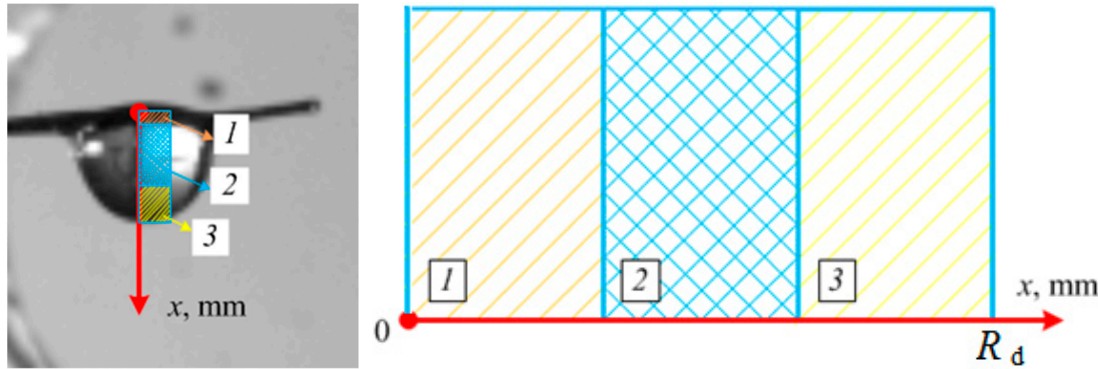

**Figure 8.** Schematic representation of the solution domain for the problem of two-component droplet heating on a holder: 1—holder, 2—droplet core (water as the first component), 3—droplet envelope (oil as the second component).

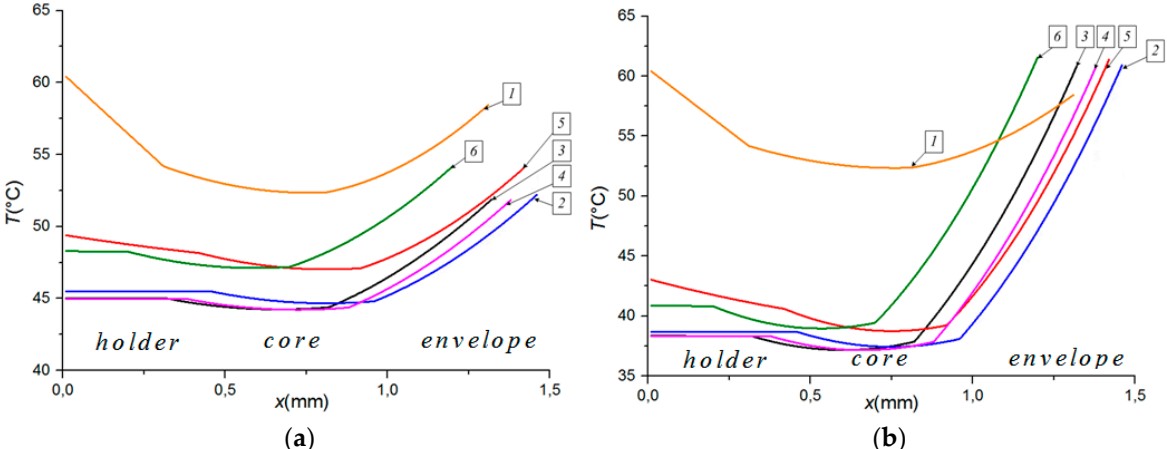

**Figure 9.** Temperature distributions (along $x$ in a holder–two-component droplet system, see Figure 8) at $T_a$ = 300 °C for 10 s (**a**) and $T_a$ = 400 °C for 5 s (**b**) using various holders: 1—phosphorus, 2—aluminum, 3—steel, 4—copper, 5—ceramics, 6—nichrome; the holder and droplet dimensions were chosen in line with the conditions of the experiments; the boundaries of areas showing the holder, core, and envelope of a droplet are not marked, since they were slightly different for each of the holders used (see Figure 1).

Theoretical plots (Figure 9) show the following: with aluminum, steel, copper, ceramic, and nichrome holders (2-6), the heat inflow to the droplet comes mostly from the flammable liquid (oil), and in the case of the phosphorus holder, from the holder (1). From this, we can conclude that the longest period of droplet heating will be provided by using a phosphorus holder, since the heat inflow comes from both the holder and the flammable liquid. Moreover, these curves (Figure 9) show that smallest heat inflow from the holder will be provided by using steel (3) and copper (4) holder materials.

In Figure 9, a significant increase in the temperature of a two-component droplet outpaces the increase in the temperature of most holders used. Before the breakup, the water–flammable liquid interface exceeded the water boiling temperature (100–120 °C). It was difficult to show such trends in one measurement system in Figure 9, since for various holders, droplets are heated and they disintegrate at various typical rates (the simulation results for water droplets are considered in research [29]). Therefore, to demonstrate the highly inhomogeneous temperature profile, Figure 9 presents the calculations for temperatures, at which a droplet remains in one piece, i.e., before explosive breakup.

In terms of practical importance, the experimental research proved that it is possible to provide adequate high-temperature liquid treatments by the explosive breakup of droplets containing various components in various proportions. In chambers used for high-temperature evaporation and the burnout of impurities, multi-component droplets swirl through high-temperature turbulent and pulsating gas media. Therefore, droplets are heated almost uniformly throughout their surface until they reach the conditions sufficient for heat removal (droplet cooling), e.g., when a droplet is fixed on a holder [15,16] or its substrates [17]. The fixation scheme of a two-component droplet on a holder, chosen in this research, is fully in line with such conditions. This scheme provides adequate evaluation of the main parameters of high-temperature liquid treatment from any impurities promoting a rapid (explosive) breakup of multi-component droplets (e.g., slurries, emulsions, and solutions).

Experiments with droplets fixed on the holders used in this study (especially the phosphorus one) make it possible to reproduce the conditions of liquid heating in high-temperature chambers. Droplets move in such chambers at almost the same velocities as the carrier medium-heated gas. The carrier medium velocities are as low as several meters per second. Therefore, liquid droplets are mostly heated by the radiative heat flux. Droplet fixation on a holder with a very low thermal diffusivity leads to a slight increase in the convective component of the flux as compared to the real-life evaporation and burnout of impurities. However, our estimates show that these deviations do not exceed 10%, and they decrease with the growing temperature of the carrier medium (Figure 9). Also, the research results can be used for development of effective approaches to the secondary atomization of droplets in fuel technologies [37–39].

## 4. Conclusions

(i) The breakup of a two-component droplet is connected with the overheating of the *water–flammable liquid* interface above the water boiling temperature (100–120 °C). Liquid surface tension forces suppress the free release of the vapor bubbles formed near the interface. When the vapor pressure in a droplet exceeded the threshold value, the droplet broke up to form a mist, aerosol, or several droplets.

(ii) When analyzing the heating times of the two-component droplets until breakup, we discovered that the disintegration times of two-component droplets are minimum when holders with a low thermal diffusivity are used ($a < 10$ m$^2$/s), and maximum when thermal diffusivity is high ($a > 80$ m$^2$/s).

(iii) In comparison with the results obtained onto a suspending thermocouple junction, under similar conditions of heat source and emulsion properties, the lifetime of the drop is close to the same order of magnitude (2–6 s).

**Author Contributions:** D.A. and M.P. conceived and designed the experiments; D.A. and O.V. performed the experiments; D.A., J.B., D.T., P.M., O.V. and M.P. analyzed the data and wrote the paper.

**Funding:** This research received no external funding.

**Acknowledgments:** Research was supported by the Russian Science Foundation (project 18–71–10002).

**Conflicts of Interest:** The authors declare no conflict of interest.

## Nomenclature and Units

| | |
|---|---|
| $a$ | thermal diffusivity, $m^2/s$ |
| $C$ | specific heat capacity, $J/(kg\cdot{}^\circ C)$ |
| $d_h$ | holder diameter, mm |
| $m$ | number of groups |
| $n$ | number of droplets in each group |
| $R_d$ | droplet radius, mm |
| $R_{d0}$ | initial two-component droplet radius, mm |
| $R_{d1}$ | droplet radius before breakup, mm |
| $R_{dn}$ | mean radius of droplets in a group, mm |
| $S$ | total area of droplet evaporation surface after breakup, $mm^2$ |
| $S_0$ | initial droplet surface area, $mm^2$ |
| $S_1$ | droplet surface area before breakup, $mm^2$ |
| $S_h$ | contact surface area of a droplet and holder surface, $m^2$ |
| $S_m$ | frontal cross-sectional area of droplet, $mm^2$ |
| $S_n$ | evaporation surface area in each droplet group, $mm^2$ |
| $T$ | temperature, $^\circ C$ |
| $T_a$ | gas flow temperature, $^\circ C$ |
| $T_d$ | temperature in a droplet, $^\circ C$ |
| $t$ | time, s |
| $U_a$ | high-temperature gas flow velocity, m/s |
| $V_d$ | drop volume, μL |
| $We$ | Weber number |
| $x$ | coordinate in a one-dimension model, mm |
| η | flammable liquid concentration, vol% |
| λ | thermal conductivity, $W/(m\cdot{}^\circ C)$ |
| ρ | density, $kg/m^3$ |
| τ | two-component droplet breakup times, s |
| $\tau_h$ | two-component droplet lifetimes, s |

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
