# Peer review of "Impact of Holder Materials on the Heating and Explosive Breakup of Two-Component Droplets"

_energies, doi:10.3390/en11123307_

Round 1

Reviewer 1 Report

This paper investigated the effect of the droplet holder material on the breaking-up of the two-component droplets during the overheating process. Several parameters including the thermal diffusivity of the holder and the concentration of the two components were found important to the breaking-up of the droplets. The paper is well-structured with sufficient references. I have the following comments for the authors to address:

1. Sometimes the authors used “two-liquid droplets” instead of “two-component droplets”. Is there a difference? If not, I would suggest using only “two-component droplets”.

2. The heating of the droplets was not described in the paper, which should be included.

3. There are many materials with low thermal diffusivity that can be used to be the droplet holder. Why did the authors choose phosphorus? Phosphorus is also flammable in air. Will it be a problem during the droplets heating?

Author Response

Please find Detailed Responses to Reviewer #1 in application.

Reviewer 2 Report

 The article Impact of holder materials on the heating and explosive breakup of two-liquid droplets is interesting and well written article concerning the experimental verification of the influence of the material used as holder of the droplet on their  explosion properties in various temperatures. This is an important problem for the  analysis of experiments as the theoretical modeling is difficult the main source of information are experimental investigations. Results from the article are valuable for interpretation of various data are are important . 

Author Response

Please find Detailed Responses to Reviewer #2 in application.

Reviewer 3 Report

This manuscript presented works on the impact of six different holder materials on the heating and explosive breakup behavior of two-liquid droplets. The authors did solid work and showed interesting results. However, the draft needs to be improved regarding writing and data discussion. Here are some comments.

In the introduction, the gap of the research is not very clear. The authors should show the different side of the proposed work and its impact on the research in this area. Also, there is another review section. It is better to put this two section together. 

Double check the figure and table number through the draft. Some of them did not match the content. For example, page 5, line148. It should be Fig.2.

In Figure 3, the authors need to mark a, b, c,...f in the figure.

Check the units through the draft and make sure they follow the journal`s requirement. An unusual unit, % wt., appears on Page 9, line 247.

In the discussion section, please make sure the statement is precise and concise. For example, Page 11, lines 314 and 315. The statement is confusing. Which holder provides the smallest impact?

Page 12, lines 348 and 349. Any data support the author`s estimation?

Author Response

Please find Detailed Responses to Reviewer #3 in application.

Reviewer 4 Report

Paper presents experimental results on the influence of support material on the heating and explosion behavior of two component droplets. Subject is very interesting, since is related to technologies that use combustion. Materials influence on heating time, minimum temperature for droplet breakup, etc… were studied. 

English is good and results are well presented. References are actual and relevant. Figures are clear. Only Table 4 could be improved reducing font size.

Author Response

Please find Detailed Responses to Reviewer #4 in application.

Round 2

Reviewer 1 Report

The authors have address the issues in the previous version of the manuscript, and I have no further comments.

Author Response

We thank the Reviewers and Editors for their helpful comments in the second review round. Please find our more detailed answers to all the questions and comments in 02_Detailed_Responses to Reviewers_1. 

Reviewer 3 Report

The authors made effort to improve the draft. However, some of the data discussion is still unclear. Here are some comments.

In the response, the authors mentioned: "the changes in the manuscript are marked in green". However, I can not find these marked content in the manuscript. Please check.

Table 1 is not well organized. The authors need to highlight the content that they want to focus. It is hard to follow the information. 

In Figure 3, "a" is still missing. Also, the other markers are not very clear. They should be more distinct. 

More work is needed to improve the discussion to make the statement precise and concise.

The conclusion and statement should be made based on data not estimation.

Author Response

We thank the Reviewers and Editors for their helpful comments in the second review round. Please find our more detailed answers to all the questions and comments in 02_Detailed_Responses to Reviewers_3. 

Round 3

Reviewer 3 Report

The authors improved the draft regarding background introduction, data analysis, and conclusion. Please double check the spelling, grammar, and format through the manuscript.